# Assessing gut microbial provisioning of essential amino acids to host in a mouse model with reconstituted gut microbiomes
Paul Ayayee [1] ✉, Gordon Custer[2], Jonathan Brent Clayton [1,3], Jeff Price[3,4], Amanda Ramer-Tait [3,4] & Thomas Larsen [5,6] ✉

Gut microbial provisioning of essential amino acids (EAAs) represents a critical but poorly understood aspect of mammalian nutrition, with direct implications for host metabolism, growth, and disease resistance. While advances in microbiome research have highlighted the potential significance of these microbial-host nutritional interactions, direct empirical evidence quantifying actual microbial contributions to host EAA supply remains surprisingly limited, particularly under controlled experimental conditions. Here, we show using stable carbon isotope analysis of six EAAs across brain, kidney, liver, and muscle tissues that germ-free mice maintained on a high-protein diet and conventionalized mice with reconstituted gut microbiomes fed a low-protein diet for twenty days exhibit no significant differences in $\delta^{13}$C-EAA values. Our results reveal no detectable microbial contribution to host EAA pools, as $\delta^{13}$C-EAA patterns remain nearly identical between treatment groups across all organs examined. Microbial profiling confirms that conventionalized mice successfully established diverse gut microbiota communities dominated by typical Firmicutes and Bacteroidetes phyla. These findings contrast with recent $\delta^{13}$C-EAA based studies that reported substantial microbial EAA contributions in wild-type mice, raising important questions about functional restoration of reconstituted gut microbiomes and underscoring the need to critically revisit experimental designs and analytical frameworks to better understand microbial nutrient provisioning dynamics.

The gut microbiome produces a range of metabolites including short-chain fatty acids (SCFAs), neurotransmitters, and essential amino acids (EAAs)[1–3] that influence host health and physiology. SCFAs, such as butyrate, acetate, and propionate, are well-documented for their roles in host energy metabolism, immune modulation, and brain function through their ability to cross the blood-brain barrier[4,5]. In contrast, the EAAs, which are critical for protein synthesis and numerous cellular functions, must be obtained from dietary sources or synthesized de novo by gut microbes for host assimilation. Unlike SCFAs, which are readily absorbed through specific transporters in the colonocytes, microbial EAAs are produced intracellularly[6] and released in the gut following bacterial lysis. These EAAs are primarily absorbed in the small intestine, particularly the jejunum[7,8]. Although active amino acid transporters exist in the colon[9], human and porcine studies show <10% of

dietary amino acids reach the large intestine, with colonic absorption contributing <5% to systemic AA pools[7,8]. Hence, understanding the extent to which microbial EAAs contribute to host amino acid pools remains complicated, given its context-dependent nature[10,11].

Studies investigating gut microbial EAA provisioning have employed a range of isotope-based approaches. Early methods relied on adding isotope labels such as $^{14}$C or $^{15}$N to trace microbial synthesis and host assimilation[10,12]. More recently, researchers have used natural isotope variability among single amino acids, especially $\delta^{13}$C values of EAAs, to track contributions of dietary and non-dietary origins of EAAs in consumer tissues[13–15]. The rationale is that, under dietary equilibrium, $\delta^{13}$C-EAA values in consumer tissues will closely match their dietary $\delta^{13}$C-EAA values with minimal isotopic fractionation and offsets, if the EAAs are sourced

[1]Department of Biology, University of Nebraska at Omaha, Omaha, NE, USA. [2]Department of Natural Sciences, University of Maryland Eastern Shore, Princess Anne, Maryland, MD, USA. [3]Nebraska Food for Health Center, University of Nebraska-Lincoln, Lincoln, NE, USA. [4]Department of Food Science and Technology, University of Nebraska-Lincoln, Lincoln, NE, USA. [5]Max Planck Institute of Geoanthropology, Jena, Germany. [6]Institute for Prehistoric and Protohistoric Archaeology, University of Kiel, Kiel, Germany. ✉e-mail: payayee@unomaha.edu; larsen@gea.mpg.de

**Fig. 1 | Carbon isotope analysis framework for detecting microbial essential amino acid contributions in animal hosts.** Conceptual scenarios based on fictional data illustrating how carbon isotope analysis of EAAs can diagnose the extent of microbial EAA supplementation in animal hosts. **A**, **C** Null hypothesis ($H_0$): No microbial EAA contribution, as indicated by host tissue $\delta^{13}$C-EAA values closely matching dietary values in both theoretical scenarios (**A**) and experimental comparisons between germ-free (GF) and conventionalized (CVZ) mice (**C**). **B**, **D** Alternative hypothesis ($H_1$): Substantial microbial EAA contribution (~25%), where host $\delta^{13}$C-EAA values are enriched and intermediate between diet and gut bacterial values (**B**), with CVZ mice exhibiting enriched $\delta^{13}$C-EAA values compared to GF mice (**D**). Lines connecting $\delta^{13}$C-EAA values illustrate source-diagnostic patterns, while dotted lines represent the dietary baseline where host $\delta^{13}$C-EAA values equal diet values, expected under full dietary equilibration with no microbial input. Created in BioRender. Larsen, T. (2025) https://BioRender.com/l14k831 and Adobe Illustrator v. 29.7.

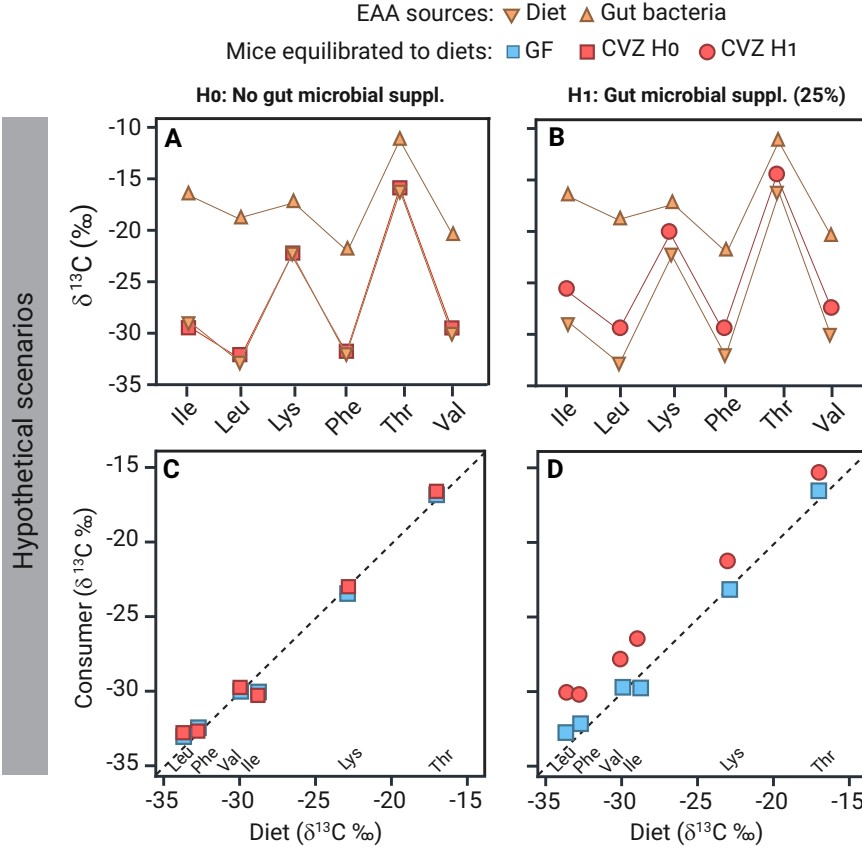

directly (Fig. 1A, C)[14]. In contrast, if the consumer also derives EAAs from gut microbes (for example bacteria), $\delta^{13}$C-EAA values in its tissues will be intermediate between those of the diet and gut bacteria (Fig. 1B, D). Evidence suggests that a diet-to-consumer offset exceeding 1‰ (the margin of analytical and biological variability)[14,16] is indicative of gut bacterial contributions to host[14,17]. The magnitude of this enrichment also depends on the $\delta^{13}$C values of individual dietary macronutrients serving as metabolic precursors for gut bacterial EAA biosynthesis. A method termed the $\delta^{13}$C-EAA fingerprinting can be used to infer the biosynthetic origins of EAAs in consumers[15,17–22]. This approach involves mean-centering the $\delta^{13}$C values of EAAs within a sample and creating a multivariate pattern, a "fingerprint", that is taxonomically diagnostic for the biosynthetic origin of the domains capable of the complete synthesis of those EAAs (bacteria, fungi, or plants)[23]. With a fingerprinting model comprising relevant biosynthetic EAA sources, the model can under natural dietary conditions assign host samples to each classifier groups (bacteria, fungi, or plants) based on $\delta^{13}$C-EAA patterns[15]. Validating and empirically demonstrating gut microbial functions in animal hosts remains a major challenge in animal-gut microbiome research, and stable isotope approaches offer the most direct means to establish causal links between gut microbiome metabolic processes and host nutritional status[23].

To translate laboratory findings into broader biological and clinical contexts, it is important to explicitly assess the putative nutritional functions of gut microbes, such as provisioning of EAAs, to host. Previous studies using $\delta^{13}$C-EAA fingerprinting in mice reported substantial gut microbial contributions to valine (~60%) and isoleucine (~40%) under low protein content (9%)[24]. These estimated contributions substantially exceed those reported in a feeding study with pigs by Torrallardona et al.[12] that used a different approach (with $^{15}$N and $^{14}$C labels), where only around 10% of the pigs' lysine requirements were provided by gut microbes. However, these previous mammalian studies did not utilize germ-free controls nor use the fingerprinting method, limiting their capacity to conclusively attribute observed isotopic offsets to microbial sources. These limitations underscore

the need for studies employing germ-free controls to definitively quantify the contributions of gut microbes to host EAA pools under controlled dietary conditions.

Our study attempts to address this limitation by comparing germ-free mice with conventionalized mice that received fecal microbiota transplants (FMT)[25,26], and directly assessing microbial EAA contributions to the host (see Fig. 2 for a schematic overview of the experimental design). This approach provides insight into the functional capacities of experimentally reconstituted gut microbiomes regarding EAA provisioning to the host[27–29]. Reconstituted microbiomes are increasingly employed as experimental models to elucidate host-microbe interactions and inform microbiome-based therapeutic strategies. Yet, whether these microbiomes fully recapitulate the functional capabilities of naturally developed communities remains unclear. Differences in colonization history, microbial community assembly dynamics, quality of diet fed to FMT transplant recipients, and functional lag times could result in incomplete or altered microbial functionality compared to naturally colonized hosts[27–31]. Thus, explicitly assessing the putative nutritional functions of reconstituted microbiomes is essential for translating laboratory findings into broader biological and clinical contexts.

In this study, we compared gut microbial EAA contributions in germ-free (GF) mice lacking a gut microbiome and conventionalized (CVZ) mice with reconstituted FMT microbiome that ostensibly function normally. Both groups were fed the same diet with one protein source, lactic acid casein, but the CVZ mice received a diet with overall lower protein content than the GF mice (10 vs. 18 kcal% protein, respectively). Given previous evidence of vertebrate gut microbial EAA provisioning under protein-restricted but not deficient diets[12,24], we hypothesized that CVZ mice would exhibit discernible differences in $\delta^{13}$C-EAA values relative to GF mice and have $\delta^{13}$C-EAA patterns indicative of microbial contributions to the host. Contrary to our hypothesis, we found no detectable microbial essential amino acid contribution to host tissues across brain, kidney, liver, and

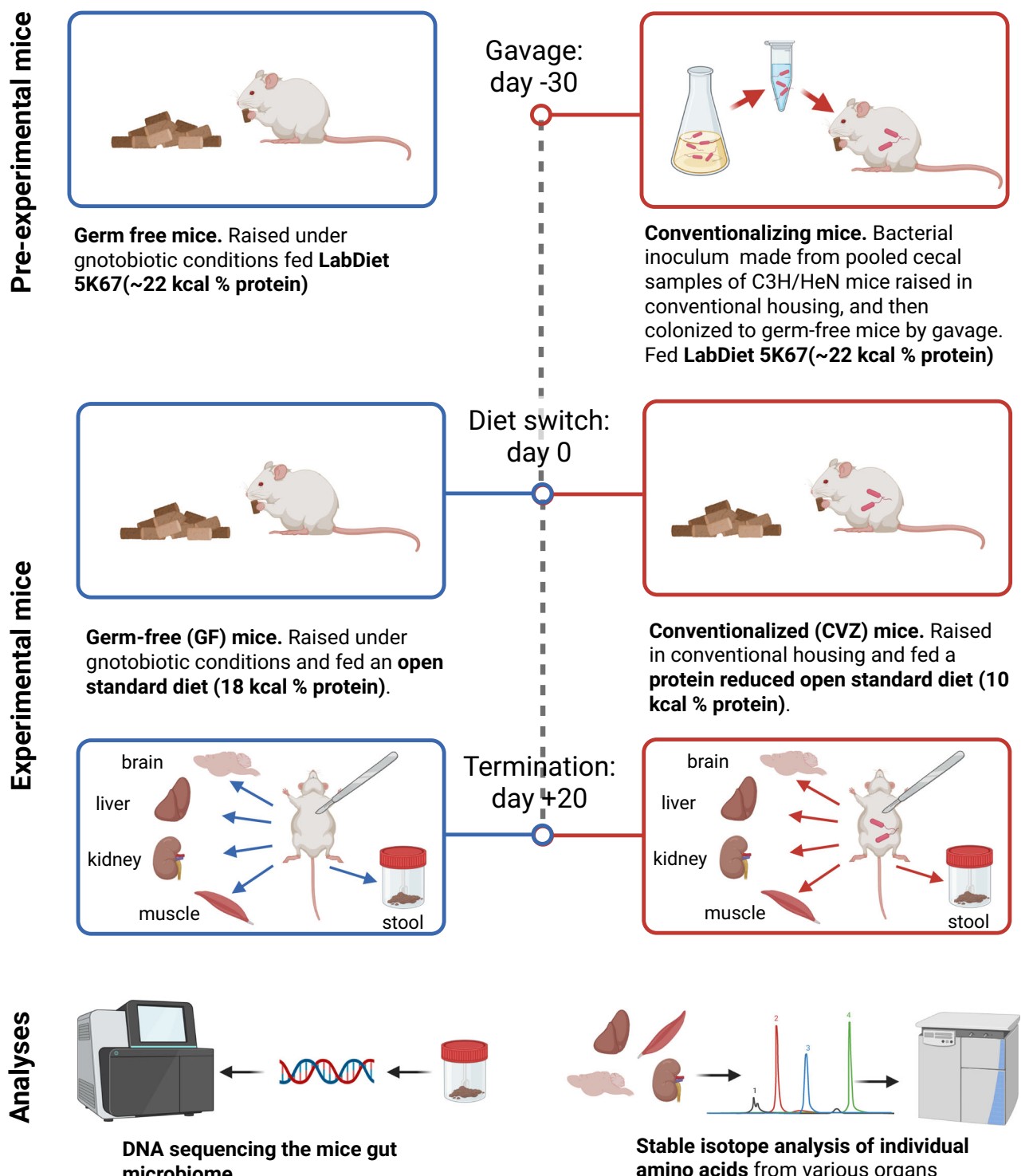

**Fig. 2 | Experimental design and analytical workflow.** Conceptual diagram illustrating the experimental design and analysis workflow used in this study. Germ-free (GF, $n = 5$) mice were raised under gnotobiotic conditions and fed an open standard diet (18 kcal% protein), while conventionalized (CVZ, $n = 5$) mice were raised in conventional housing and fed a protein-reduced open standard diet (10 kcal% protein). Both groups were terminated 20 days after the diet switch, and brain, liver, kidney, muscle, were dissected for stable isotope analysis of individual amino acids and stool samples were taken for DNA sequencing of the gut microbiome. Created in BioRender. Larsen, T. (2025) https://BioRender.com/l14k831 and Adobe Illustrator v. 29.7.

muscle. $\delta^{13}C$-EAA values remained nearly identical between germ-free and conventionalized mice despite successful microbiome reconstitution according to the gut microbiome composition. These findings suggest that reconstituted gut microbiomes may have limited functional capacity for EAA provisioning and highlight important considerations for interpreting microbial nutritional contributions in experimental settings.

## Results

### Gut microbial EAA supplementation

The $\delta^{13}C$-EAA values of organs from GF and CVZ mice were plotted against the Open Standard diet to visualize tissue specific offsets (Figs. 3 and 4A, B). Welch's $t$ tests revealed a significant difference in muscle tissue $\delta^{13}C$-EAA values between CVZ ($3.65 \pm 0.25‰$) and GF mice ($4.24 \pm 0.31‰$;

**Fig. 3 | Tissue-specific carbon isotope offsets between treatment groups.** Tissue-specific $\Delta\delta^{13}$C-EAA (Ile, Leu, Lys, Phe, Thr, Val) offsets between conventionalized (CVZ, blue) and germ-free (GF, red) mice. Split violin plots show distributions of average tissue-diet $\delta^{13}$C offsets across EAAs, with individual mouse means represented by points. Violin width reflects probability density. Source data: Supplementary Table 2.

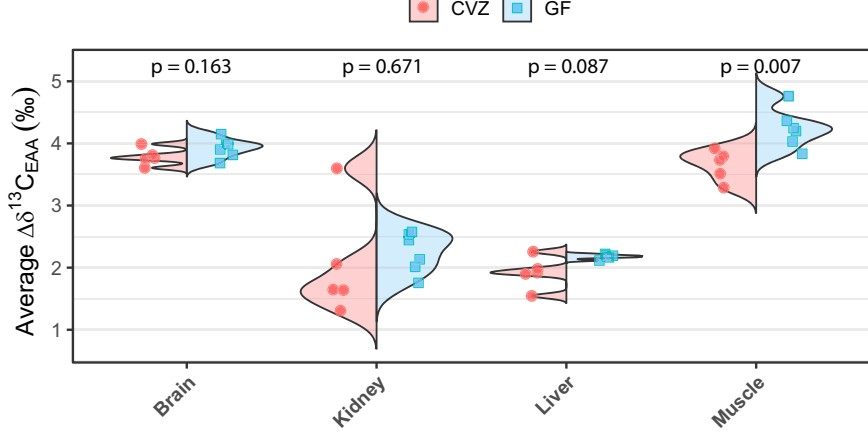

$p = 0.007$), but brain, kidney, and liver showed no treatment effects ($p > 0.05$) (Fig. 3). Organ $\delta^{13}$C-EAA values were consistently enriched relative to dietary values, with the highest enrichment observed in the liver, followed by kidney, muscle, and brain (Table 1). Among the EAAs, threonine (Thr) was the most enriched, whereas lysine (Lys) was the least (Fig. 4A). Multivariate analysis demonstrated significant differences in $\delta^{13}$C-EAA values across organs ($F_{9, 266} = 154.1$, $P < 0.0001$) and EAAs ($F_{5, 270} = 118$, $P < 0.0001$), with a significant interaction between dietary groups and amino acids ($F_{45, 230} = 12.45, P < 0.0001$) (Table 1). We found no treatment effects for non-essential amino acids except for the kidney at $P = 0.038$ (Fig. 4B).

Within the $\delta^{13}$C-EAA fingerprinting framework, the probabilistic distributions yielded a median Bhattacharyya coefficient (BC) value of 0.941, with a range from 0.716 to 0.997 across 46 observations. The interquartile range (IQR) was 0.917 (Q25) to 0.962 (Q75), indicating a high degree of overlap between GF and CVZ groups. These BC metric results suggest that the $\delta^{13}$C-EAA patterns for GF and CVZ organs are highly similar (Fig. 5A, B).

## Gut microbiome composition

Standard read quality-processing, amplicon sequence variants (ASV) determination, and curation (removal of ASV assigned mitochondrion, chloroplast, and unassigned at the domain level) yielded 282 ASVs distributed across 10 samples. Rarefaction at 1400 reads per sample resulted in a final ASV tally 171, with rarefaction curves indicating sufficient coverage across samples (Supplementary Fig. 1). Initial cursory analyses indicated the dominance of contaminating Streptococcaceae (presumably from the diet) in GF-mice samples. Subsequent removal of Streptococcaceae ASVs resulted in unacceptably low counts for the remaining ASVs in GF mice samples, confirming that the GF mice maintained their germ-free status throughout the twenty-day experimental period, in contrast to the CVZ mice. Thus, GF-mice samples were removed, and downstream analyses were performed for only individual CVZ mice.

Microbial richness (Chao1; $86.6 \pm 4.88$, and observed species; $75.8 \pm 3.45$, mean ± S.E) and evenness (Shannon's index; $3.86 \pm 0.20$) estimates did not vary significantly among CVZ mice ($P = 0.41$ for all estimates). Lachnospiraceae (Phylum Firmicutes; abundant genus, Unassigned Lachnospiraceae) (32.92%), Muribaculaceae (Phylum Bacteroidetes; genus Unassigned Muribaculaceae) (34.14%), Bacteroidaceae (Phylum Bacteroidetes; genus *Bacteroides*) (10.18%), Erysipelatoclostridiaceae (Phylum Firmicutes; genus Erysipelatoclostridium) (8.37%), and Rikenellaceae (Phylum Bacteroidetes; genus *Alistipes*) (4.67%) were identified as dominant taxa in CVZ microbiomes (Fig. 6A, B).

## Discussion

We examined whether CVZ mice with reconstituted gut microbiomes, when fed a low-protein diet, provision EAAs to the host. The study was designed so that the overall functional outcome, namely, the host's acquisition of microbial EAAs, could be evaluated independently of which microbial species were present in the gut. By using $\delta^{13}$C-EAA fingerprinting to trace the biosynthetic EAA origins, we measured the overall functional effect of microbial activity as it occurred, rather than relying on predictions based on 16S rRNA community composition. Despite determining that the CVZ gut microbiome was dominated by Firmicutes and Bacteroidetes, consistent with typical and healthy mouse gut ecosystems, we found no evidence of significant microbial EAA provisioning after a twenty-day experimental feeding period, thus failing to support our initial hypothesis. While we detected significant differences between the two treatments for muscle tissues, the averaged 0.6‰ offset falls below the typical and widely used detection threshold (1‰) for microbial EAA provisioning. Furthermore, the GF group was more $^{13}$C-enriched than the CVZ group, contrary to expectations from microbial EAA provisioning. Also, our $\delta^{13}$C-EAA fingerprinting analysis revealed nearly complete overlap between GF and CVZ organs in $\delta^{13}$C pattern space, indicating that the predictive ordinations relative to the training data were almost identical for both groups across all organs. These findings contrast with previous studies reporting substantial microbial contributions to host EAA pools in mice and other mammals with natural gut microbiomes[12,24]. The discrepancy between our findings and previous studies may be attributed to several key methodological differences.

First, our use of germ-free controls allowed for direct assessment of microbial contributions by treating the microbiome as the sole variable. This approach enabled direct attribution of any isotopic offset to microbial activity, rather than to dietary or metabolic confounders. By comparing GF mice lacking gut microbiota with CVZ mice with reconstituted gut microbiomes, any differences in $^{13}$C-EAA values could be directly attributed to microbial provisioning, eliminating confounding factors present when using mice with established microbiomes only. This approach provides direct empirical evidence of microbe-derived metabolite incorporation into host tissues, in contrast to metagenomic or metatranscriptomic methods, which can infer only functional potential rather than actual metabolic transfer.

Second, the use of conventionalized mice enabled regulation of the inoculating gut microbiota for assessment of this function. Our CVZ mice exhibited a typical gut microbiome composition dominated by Firmicutes (Clostridia) and Bacteroidetes (Bacteroidota) with *Lachnospiraceae* and *Muribaculaceae*, respectively, as dominant families[32,33]. Interestingly, despite most *Lachnospiraceae* species being prototrophic for all amino acids[33,34] (able to synthesize them), their prevalence did not translate to detectable EAA provisioning to the host in our study. However, their metabolic specialization in carbohydrate fermentation means their contribution to host EAA pools was likely minimal. Similarly, *Muribaculaceae*, which showed significant presence in our CVZ mice, have been reported to upregulate genes for carbohydrate metabolism[33]. Our findings with reconstituted gut

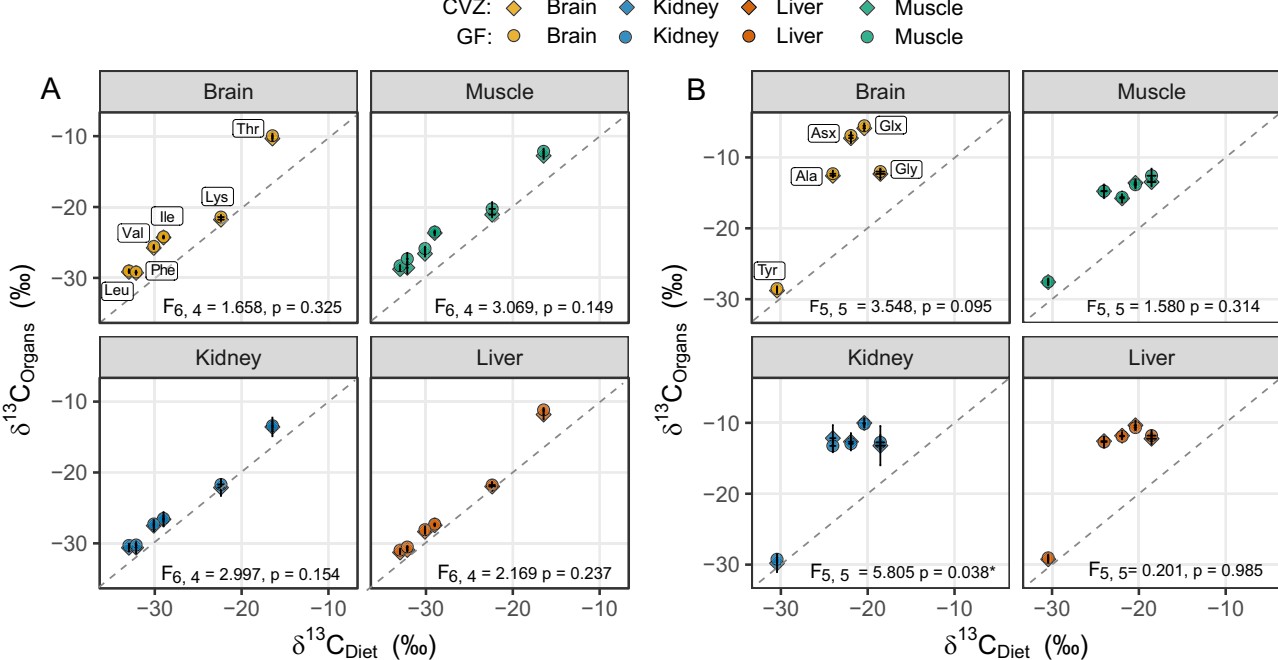

**Fig. 4 | Relationship between organ and dietary carbon isotope values.** Scatter plot illustrating the relationship between the amino acid $\delta^{13}C$ values of mouse organs and diets, 20 days after switching from LabDiet 5K67 to an Open Standard Diet. The Open Standard Diet contained 10 kcal% protein for conventionalized (CVZ) mice and 18 kcal% protein for germ-free (GF) mice. **A** Essential amino acids. **B** Non-essential amino acids. For both panels, the x-axis represents the mean $\delta^{13}C$ value of the Open Standard Diet proteins ($n = 2$) and the y-axis represents the mean $\delta^{13}C$ value of CVZ ($n = 5$) or GF ($n = 6$) organs. The p-values indicate Pillai's Trace from a one-way multivariate analysis of variance. There are no treatment effects except for the kidney non-essential amino acids at $p < 0.05$. Source data: Supplementary Table 2.

### Table 1 | Mean ± standard deviation (SD) values of $\delta^{13}C$-EAA values for germ-free (GF) and conventionalized (CVZ) mice across brain, kidney, liver, and muscle tissues

| | Ile | Leu | Lys | Phe | Thr | Val | Avg |
|---|---|---|---|---|---|---|---|
| **Brain** | | | | | | | |
| GF | 4.8 ± 0.2 | 4.0 ± 0.2 | 0.7 ± 0.3 | 2.9 ± 0.2 | 6.5 ± 0.3 | 4.5 ± 0.2 | 3.9 ± 1.8 |
| CVZ | 4.7 ± 0.1 | 3.8 ± 0.1 | 0.8 ± 0.2 | 2.9 ± 0.2 | 6.1 ± 0.3 | 4.4 ± 0.2 | 3.8 ± 1.7 |
| *P* value | 0.154 | 0.190 | 0.521 | 0.829 | 0.053 | 0.149 | NA |
| Sign. level | NS | NS | NS | NS | NS | NS | NA |
| **Kidney** | | | | | | | |
| GF | 2.5 ± 0.4 | 2.7 ± 0.1 | 0.5 ± 0.4 | 1.9 ± 0.1 | 3.1 ± 0.6 | 2.8 ± 0.5 | 2.2 ± 1.0 |
| CVZ | 2.4 ± 1.0 | 2.4 ± 0.5 | 0.5 ± 1.2 | 1.6 ± 0.8 | 2.9 ± 1.3 | 2.6 ± 0.8 | 2.1 ± 1.2 |
| *P* value | 0.736 | 0.231 | 0.968 | 0.465 | 0.786 | 0.625 | NA |
| Sign. level | NS | NS | NS | NS | NS | NS | NA |
| **Liver** | | | | | | | |
| GF | 1.7 ± 0.1 | 2 ± 0.2 | 0.4 ± 0.4 | 1.6 ± 0.1 | 5.3 ± 0.3 | 2.0 ± 0.5 | 2.2 ± 1.5 |
| CVZ | 1.5 ± 0.1 | 1.7 ± 0.2 | 0.6 ± 0.6 | 1.3 ± 0.3 | 4.6 ± 0.7 | 1.8 ± 0.3 | 1.9 ± 1.3 |
| *P* value | *0.027* | *0.037* | 0.439 | 0.050 | 0.099 | 0.323 | NA |
| Sign. level | * | * | NS | NS | NS | NS | NA |
| **Muscle** | | | | | | | |
| GF | 5.5 ± 0.1 | 4.7 ± 0.1 | 1.9 ± 0.9 | 4.8 ± 0.8 | 4.3 ± 0.4 | 4.2 ± 0.3 | 4.5 ± 1.2 |
| CVZ | 5.3 ± 0.3 | 4.2 ± 0.2 | 1.5 ± 0.4 | 3.6 ± 0.8 | 3.7 ± 0.5 | 3.5 ± 0.4 | 3.7 ± 1.2 |
| *P* value | 0.296 | *0.005* | 0.417 | *0.044* | *0.038* | *0.013* | NA |
| Sign. level | NS | ** | NS | * | * | * | NA |

*P*-values were calculated using Welch's *t* test. Significance levels are indicated as follows: NS (not significant), *(*P* < 0.05), **(*P* < 0.01), ***(*P* < 0.001). NA signifies not applicable. Essential amino acids were *Ile* isoleucine, *Leu* leucine, *Lys* lysine, *Phe* phenylalanine, *Thr* threonine, *Val* valine

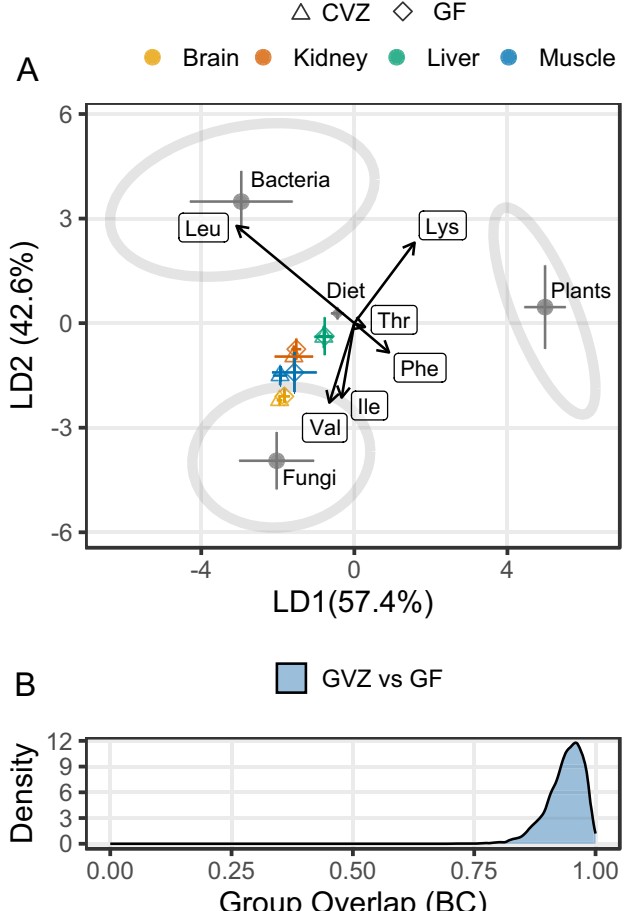

**Fig. 5 | Linear discriminant analysis of carbon isotope patterns. A** Linear discriminant function analysis (LDA) plot based on $\delta^{13}$C-EAA from GF-fed (N = 5) and CVZ (N = 5) and three classifier groups [fungi (n = 9), bacteria (n = 12) and plants (n = 11)]. The shaded ellipses signify the 95% confidence limits for each classifier group. **B** Bhattacharyya coefficients (BC) quantifying group pair overlaps based on $\delta^{13}$C-EAA values, where BC = 0 indicates no overlap and BC = 1 indicates identical distributions. The median BC value for GF and CVZ groups was 0.941, with an interquartile range of 0.917–0.962, reflecting a high degree of similarity in probabilistic distributions. Essential amino acids were Ile isoleucine, Leu leucine, Lys lysine, Phe phenylalanine, Thr threonine, Val valine. Source mice and feed data: Supplementary Table 2. Source $\delta^{13}$C-EAA training data: https://doi.org/10.1371/journal.pone.0073441.s004.

microbiomes align with traditional understanding of mammalian digestive physiology. Protein digestion and amino acid absorption occur primarily in the small intestine, whilst the colon has limited absorption capacity despite being where most gut microbes reside[12,35]. Taken together, our community compositional findings using fecal samples indicate that the dominant fecal bacterial taxa (Lachnospiraceae and Muribaculaceae) in the CVZ mice may be compositionally similar to those in the large intestines (whose community composition is similar to fecal samples)[36] and specialize in carbohydrate fermentation and SCFA production, underscored possibly by loosely connected syntrophic EAA biosynthesis pathways[29,34]. While our low-protein diet (10 kcal% protein) theoretically created conditions favorable for microbial EAA provisioning, we were unable to detect significant contributions under this condition.

Finally, our methodological approach using a single protein source (lactic acid casein) with varying protein content (10 vs 18 kcal% protein for CVZ and GF mice, respectively) in the OpenStandard diets differs from other studies that employed diets with multiple protein sources and a wider range of protein content (9–40%)[24]. By using a single, well-defined protein

source, our approach minimizes potential discrepancies between consumed and assimilated dietary proteins and reduces biases related to dietary memory effects[36,37]. Although we used OpenStandard diets with a single protein source (lactic acid casein) during the experimental feeding period, the CVZ mice were initially maintained on a complex diet (LabDiet 5K67; ~22 kcal% protein from multiple sources) during the thirty-day microbiome re-establishment period post-FMT (Fig. 2). This protocol, designed to ensure mouse survival, introduced dietary complexity through the transitioning from a multi-source to a single-source protein diet, which in turn may have influenced microbial adaptation and EAA synthesis capacity[38]. Although our methodology and findings differ from previous studies, our results provide guidance for refining experimental approaches and interpretations in vertebrate host–microbiome research. First, we recognize that dietary context during microbiome establishment may critically influence the functional capacity of reconstituted gut microbiomes[31,39–44]. The thirty-day establishment phase on a standard diet (LabDiet 5K67) prior to experimental feeding could have shaped a microbiome optimized for that dietary regime. This optimization may have potentially limited EAA biosynthetic functionality under subsequent low-protein conditions. Future studies maintaining consistent experimental diets throughout establishment may yield microbiomes with enhanced functional alignment to dietary challenges.

Second, our findings underscore the need for further research into the functional capacity of reconstituted gut microbiomes. Besides using meta-transcriptomics to assess expression of microbial EAA biosynthetic pathways, comparative studies with both reconstituted and naturally established gut microbiomes would provide valuable insights into the functional equivalence of both reconstituted and naturally established gut microbiomes, and the potential ecological factors that enable or constrain microbial EAA provisioning in mammalian hosts. However, it is important to note that the methodology used in this study comparing $^{13}$C-enrichment between GF and CVZ mice relative to their diets cannot be directly implemented with specimens having naturally established gut microbiomes, since establishing a true germ-free control group is not feasible[45,46]. Comparative studies with positive controls would instead need to rely on alternative approaches such as isotope tracer additions (e.g., $^{15}$N and $^{14}$C labels used by Torrallardona et al. in pigs[12]) or $\delta^{13}$C-EAA fingerprinting, with appropriate microbial end members to quantify potential microbial EAA contributions to host[23]. Recent advances in stable isotope resolved metabolomics (SIRM) using $^{13}$C-labeled dietary fibers like inulin could also provide valuable insights by tracking the dynamic flow of microbial metabolites including amino acids, to host tissues across multiple organs[5].

Finally, the absence of detectable microbial EAA provisioning in our CVZ mice raises important questions about the functional restoration of reconstituted gut microbiomes and the suitability of current techniques and experimental designs for assessing microbial functions. Although our study focused specifically on EAA provisioning, these findings have broader implications for understanding how gut microbiome reconstitution, whether after perturbation or transplantation, affects host nutrition. The gut microbiome is highly sensitive to factors such as diet, antibiotics, and environmental stressors, which can induce lasting changes in community composition and metabolic capacity, with significant consequences for host metabolism and nutrient acquisition[31,47,48]. Notably, previous research has shown that diet-induced extinctions in the gut microbiota can persist across generations, resulting in the permanent loss of specific microbial functions[49,50]. In our context, the lack of detectable EAA provisioning may reflect incomplete functional restoration in the CVZ microbiome, rather than an inherent inability of gut microbes to contribute EAAs to their hosts[50]. For example, studies with gnotobiotic mice have demonstrated that colonization with microbiota from undernourished children can impair growth and metabolism[51]. Our findings therefore highlight the need for a more detailed, context-specific understanding of how experimental design parameters, including the history and duration of microbiome establishment, influence the restoration of complex metabolic functions in reconstituted communities.

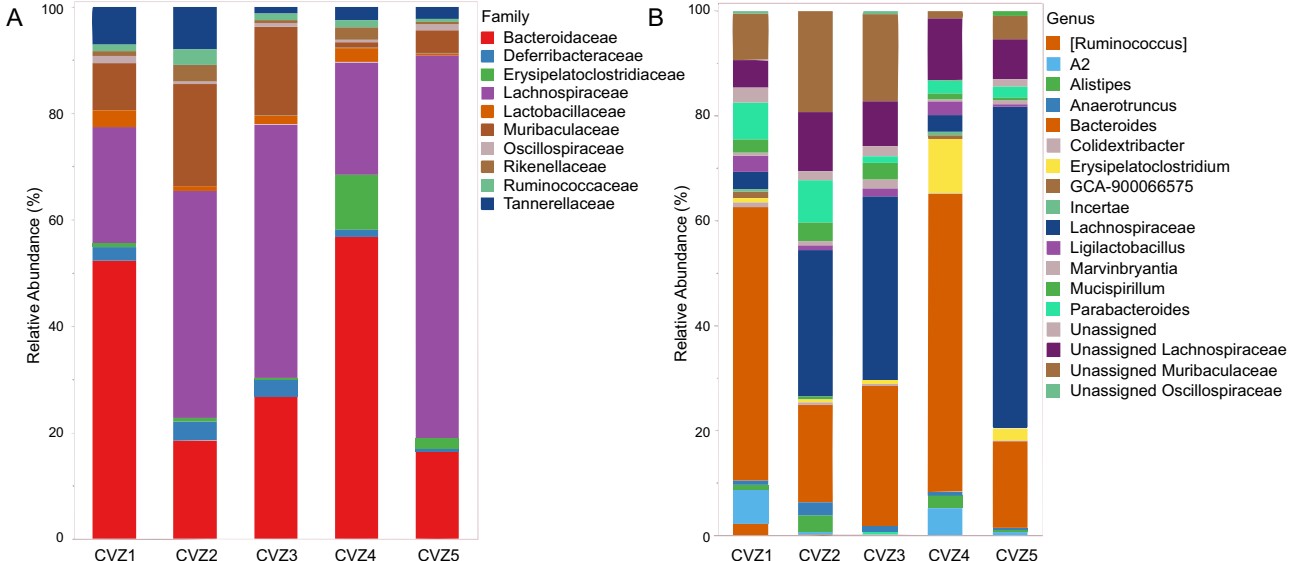

**Fig. 6 | Bacterial community composition in conventionalized mice.** The relative abundances of the most abundant bacterial taxa in the fecal microbiomes of conventionalized (CVZ) mice at the end of the study at the **A** family level and **B** genus level. Source data for relative abundances: Supplementary Data 1.xlsx.

## Conclusion

In conclusion, despite being kept on a low-protein diet for 20 days, our murine model revealed no evidence of gut microbial EAA provisioning in conventionalized mice. This unexpected finding raises important questions about the establishment and functional capacity of reconstituted gut microbiomes and the potential impacts of microbiome perturbations on host nutrition. The lack of detectable EAA provisioning across multiple organs with varying turnover rates suggests that reconstituted microbiomes need to be studied in a context-dependent manner to fully assess their functionality relative to native microbial communities. This sets the stage for more comprehensive studies to elucidate the complexities of gut microbial functions, the microbial dimensions of the nutritional ecology of omnivorous mammals, and our understanding of host-microbe interactions and the development of microbiome-focused therapies.

## Methods

### Experimental design

We used natural variations in stable carbon isotope ratios of essential amino acids (EAAs) $\delta^{13}$C-EAA to examine whether gut microbiota supply EAAs to mice (Fig. 1). Ten initial three-week-old female germ-free C57BL/6 (B6) mice ($n = 10$) were reared in flexible film isolators and maintained under standard gnotobiotic conditions by the Nebraska Gnotobiotic Mouse Program (NGMP)[52]. The cages featured flooring made of wire mesh, which allowed waste to drop below and out of reach of the animals. Half of the GF mice were transferred to individually ventilated cages and conventionalized ($n = 5$) via oral gavage with 100 μL of a conventional in-facility mouse microbiota diluted at 1:10 (w/v; grams of ceca per milliliters of PBS plus 10% glycerol)[52]. The conventional mouse microbiome was obtained initially from pooled cecal samples of C3H/HeN mice raised in conventional housing at the University of Nebraska-Lincoln[34,52]. Gavaged mice were maintained for 30 days to ensure establishment of gut microbiota before the start of the experiment. The remaining GF mice ($n = 5$) were maintained in flexible film isolators. The germ-free status of these mice was confirmed with culture-dependent approaches following standard protocols in the facility[32,52]. Before the experiment, all mice were fed an autoclaved diet (LabDiet 5K67) containing ~22 kcal% protein (Ingredient list in Supplementary Table 1). At the start of the experiment, both the GF and CVZ mice switched to an irradiated OpenStandard Diet (Research Diets, Inc.) with lactic acid casein (SureProtein™ LACTIC CASEIN 720) from grass-fed New

**Table 2 | Ingredients and macromolecular composition of purified diets OpenStandard Diet (Research Diets, Inc.) fed to GF and CVZ mice**

| Macronutrients | GF diet | | CVZ diet | |
|---|---|---|---|---|
| Ingredients | g | kcal% | g | kcal% |
| ***Protein (total)*** | *16.5* | *18* | *9.2* | *10* |
| Casein | 16.3 | | 9.1 | |
| L-Cystine | 0.2 | | 0.1 | |
| Carbohydrate (total) | 61.7 | 66 | 69.9 | 74 |
| Corn Starch | 36.2 | | 42.8 | |
| Maltodextrin | 10.4 | | 12 | |
| Dextrose | 14.2 | | 14.4 | |
| Sucrose | 0.9 | | 0.8 | |
| Fat (total) | 6.6 | 16 | 6.7 | 16 |
| Soybean Oil | 6.6 | | 6.7 | |
| Fiber (total) | 9.3 | | 9.4 | 0 |
| Cellulose | 6.98 | | 7.05 | |
| Inulin | 2.32 | | 2.35 | |

All the other components of the diets remain relatively unchanged, except for overall protein content

Zealand cows as the protein source (Table 2). The GF mice were fed an OpenStandard diet with 18 kcal% protein (D11112201), and the CVZ mice were fed the same OpenStandard diet but with reduced protein (10 kcal% protein, D22010704-1.5 V). The protein level used for the CVZ mice is at the lower end of the protein range (9–40 kcal% protein) used by Newsome et al.[24] in their study.

We selected a twenty-day feeding period for both groups, informed by published data on carbon turnover rates in mice[52–54]. We chose a multiorgan approach analyzing the liver, kidneys, muscle, and brain, enabling the detection of potential differences in tissue-specific microbial EAA provisioning or EAA provisioning that might be dependent on metabolic activity. Published data on mouse carbon turnover in single amino acids have shown a replacement of 75%, 60%, 45%, and 30% in the liver, kidneys, muscle, and brain, respectively, within a twenty-day period[55]. Twenty days after the diet switch, mice were euthanized, and organs (liver, muscle, kidney, and brain)

were collected. Samples were bio-banked at -80 °C until further analyses. Putative gut microbial EAA provisioning to the host was investigated by comparing δ13C-EAA values between CVZ and GF mice. Significantly higher δ13C-EAA in CVZ than GF organs would indicate that mice had acquired EAA synthesized by their microbiome (non-dietary sources). Hence, our null hypothesis ($H_0$) posits that the gut microbiota does not contribute EAAs to the host, evidenced by an overlap in δ13C-EAA values between GF and CVZ mice. The alternative hypothesis ($H_1$) posits that the gut microbiota does contribute to EAAs, resulting in higher δ13C-EAA values in CVZ than in GF mice. We subsequently used the δ13C-EAA fingerprinting method to aid in identifying putative gut microbial EAA origins with the expectation that the dietary protein source, lactic acid casein, would classify between bacteria and plants[15]. This study received ethical approval from the Institutional Animal Care and Use Committee at the University of Nebraska-Lincoln (protocol # 2126), and we have complied with all relevant ethical regulations for animal use.

## Amino acid δ13C measurements from sampled tissues

Before determining δ13C-EAA values, liver, muscle, kidney, and brain samples from both CVZ and GF mice ($n = 5$, of each organ) were lyophilized under vacuum at $-80$ °C for 48 h. Samples ($n = 42$, four organs from 5 GF and 5 CVZ mice, two low and high protein diet samples) were then pulverized and submitted for compound-specific amino acid stable isotope analysis at the Stable Isotope Facility at the University of California, Davis (Davis, CA, USA). Briefly, all samples were first acid-hydrolyzed for 70 min under a $N_2$-gas headspace in 6 M HCl at 150 °C. Samples were then derivatized as *N*-acetyl methyl esters via methoxy carbonylation-esterification (NACME)[56,57]. Subsequently, derivatized samples were injected into a splitless liner at 260 °C and separated on an Agilent DB-35 column (60 m × 0.32 mm ID × 1.5 µm film thickness) at a flow rate of 2 mL/min under the following temperature program: 70 °C (hold 2 min); 140 °C (15 °C/min, hold 4 min); 240 °C (12 °C/min, hold 5 min); and 255 °C (8 °C/min, hold 35 min). Compound-specific isotope 13C-amino acid analysis was carried out using a Thermo Trace GC 1310 (GC; Thermo Fisher Scientific, Waltham, MA, USA) coupled to a Delta V Advantage isotope ratio mass spectrometer via the GC IsoLink II (Thermo Electron, Bremen, Germany)[57]. All samples were analyzed in duplicates. Replicates of the quality control and assurance reference materials were measured after every five samples. Standard exogenous carbon addition, kinetic isotope effects from derivatization reagents procedures, and normalization to the international reference for δ13C VPDB, as well as a calibrated amino acid mixture, UCD AA3, and multiple natural materials, were carried out as per facility practices[57]. δ13C-EAA data were obtained for six essential amino acids: isoleucine, leucine, lysine, phenylalanine, threonine, and valine across all 42 samples run in duplicates. The mean standard deviation for all samples was ± 0.2‰, well below the established quality control value of ±0.4‰. Final accuracy, as determined by the mean absolute difference in the measured and known δ13C-EAA values of EAAs from a quality assessment mixture of amino acids, was within ±0.5‰.

## DNA extraction and microbiome sequence generation

To ascertain the establishment of a normal reconstituted gut microbiome in the CVZ mice and that GF mice remained germ-free, DNA was extracted from fecal samples of individual mice ($n = 5$ each of GF and CVZ) after the experimental period. Briefly, ~0.25 g of all 10 DNA samples were extracted using the PowerSoil® DNA Isolation Kit (Qiagen, Germantown, MD, USA) according to the manufacturer's directions. Samples were submitted for high-throughput paired-end Illumina MiSeq library preparation and sequencing at the University of Nebraska Medical Center Genomics Core. Briefly, a limited-cycle PCR reaction was performed on each sample to create amplicons of the V4 (515-F; 5'-GTGCCAGCMGCCGCGGTAA-3') and V5 (907-R; 5'-CCGTCAATTCMTTTRAGTTT-3') variable region[58] with overhang adapters using the Nextera XT DNA Library Preparation Kit

(Illumina, San Diego, CA, USA). The resulting libraries were validated using the Agilent BioAnalyzer 2100 DNA 1000 chip (Agilent), and DNA was quantified using Qubit 3.0 (Qubit™, Thermofisher). The libraries were loaded into the Illumina MiSeq at 20 pM to generate 300 bp paired ends with the 600-cycle kit v3 (Illumina, San Diego, CA, USA). For quality control, the libraries were spiked with 20% PhiX (a bacteriophage) to assess and confirm sequencing efficiency[59].

## Statistics and reproducibility

**Amino acid δ13C analyses.** Amino acid δ13C analyses: All six essential amino acid data from all sampled organs ($n = 42$) were normalized to their respective dietary δ13C-EAA values before statistical analysis. We calculated the δ13C offset as $\Delta\delta^{13}C = \delta^{13}C_{Consumer\ EAA} - \delta^{13}C_{Dietary\ EAA}$[15] for each essential amino acid from each organ. To compare enrichment patterns between germ-free (GF) and conventionalized (CVZ) mice, Welch's *t* tests were applied, enabling robust comparison while accounting for heteroscedasticity between groups. Exact p-values quantify the significance of the observed differences for each organ–EAA pairing. Biosynthetic origins of EAAs in dietary casein and consumer tissues were inferred using linear discriminant function analysis (LDA), trained on δ13C-EAA reference profiles from bacteria, fungi, and plants[60] after inter-laboratory calibration for analytical consistency. The LDA was implemented using the MASS package in R, with model performance assessed by leave-one-out cross-validation. Group separation was further evaluated with pairwise Bhattacharyya coefficients (BCs)[61] on the LDA-transformed data, where values close to 1 indicate high overlap and 0 indicates distinct distributions. Data suitability for multivariate analyses was determined by Fligner–Killeen tests of variance homogeneity and by Q–Q plots for normality. All statistical analyses were performed in R v4.2.2 (http://www.R-project.org).

**Microbiome analyses.** Before processing, primers and adapters were removed using Cutadapt (v. 3.7), and acquired reads were processed with the DADA2 package in R version 1.21[62,63]. Chimeras were removed using the remove chimeraDenovo function, with method = "consensus." The final sequence table was then trimmed to include only sequences with read lengths ranging from 367 to 375 base pairs. The Silva V138 database was used to assign taxonomy[64], and the final ASV table, sample metadata, and taxonomic assignments were imported to Phyloseq[65] for downstream processing in R. The ASV count table and ASV taxonomy files were also combined and formatted for comparative analyses in Microbiome Analyst[66] with codes deposited in the MicrobiomeAnalyst GitHub repository (https://github.com/xia-lab/MicrobiomeAnalystR)[67]. Before downstream statistical analyses, all non-bacterial reads and those assigned to Eukaryota, chloroplasts or mitochondria were removed. Diversity analyses were performed only for the CVZ mice group, following confirmation of limited to no microbial presence/diversity in the GF mice. Differential abundance across each CVZ mouse was calculated using the group-significance.py command in QIIME[68], and relative abundance plots at the family and genus level were made in JMP Pro 17 (JMP® Pro, Version 17.0.0 (2022) SAS Institute Inc., Cary, NC, 1989-2022).

## Reporting summary

Further information on research design is available in the Nature Portfolio Reporting Summary linked to this article.

## Data availability

Raw sequence reads were deposited into the NCBI SRA under BioProject PRJNA927293. Source δ13C-AA data from mice and feed used in Figs. 3, 4, and 5a can be found in Supplementary Table 2. The Fig. 5a δ13C-EAA training data can be downloaded from: https://doi.org/10.1371/journal.pone.0073441.s004[60]. See Supplementary Table 3 for the abundances of the 30 most abundant bacterial taxa in the fecal microbiome of CVZ mice, "Supplementary Data 1.xlsx" for the relative abundance data, and "Supplementary Data 2.xlsx" for the rarefaction plot data.

## Code availability

All codes used for processing raw amplicon reads and statistical analyses are at Zenodo[69] (https://doi.org/10.5281/zenodo.16914341).

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

## Acknowledgements

This study was funded by the start-up grant provided by the University of Nebraska, Omaha, to Dr. Paul Ayayee. We acknowledge the Stable Isotope Facility at the University of California Davis for assistance with stable isotope analyses. This work was also completed using the University of Nebraska DNA Sequencing Core, which receives partial support from the National Institute for General Medical Science (NIGMS) INBRE-P20GM103427-19 grant and the Fred & Pamela Buffett Cancer Center Support Grant-P30 CA036727. Research reported in this publication was supported by the Office of The Director, National Institutes of Health of the National Institutes of Health under Award Number K01OD030514, awarded to J.B.C. T.L. was supported by the Max Planck Society. We acknowledge the assistance of various undergraduate lab assistants in the execution of the study, notably Rozlyn Olson and Kiera Nelson.

## Author contributions

P.A.A., T.L., J.P., J.B.C., and A.R.T. conceived and designed the study. J.P. and A.R.T. initiated the mice housekeeping. P.A.A. prepared samples for processing, and P.A.A., T.L., and G.C. analyzed data. P.A.A., T.L., J.P., J.B.C., G.C., and A.R.T. wrote the paper. All authors gave comments and final approval for publication.

## Funding

## Competing interests

The authors declare no competing interests.
