## [Transparent Peer review file · Communications Biology]

Assessing gut microbial provisioning of essential amino acids to host in a mouse model with reconstituted gut microbiomes

Corresponding Author: Dr Thomas Larsen

Version 0:

Reviewer comments:

Reviewer #1

(Remarks to the Author)

Summary

The goal of this work was to determine the microbial contribution of EAA to the host and distribution across several host tissues by comparing conventionalized mice on a low protein diet and germ free (GF mice) on a high protein diet. Microbe-derived EAAs among host tissues or that differed from GF mice were not detected. The authors correctly point out that further studies utilizing varied dietary protein levels and additional analytical and -omic techniques, for example, are needed.

Major Comments

1. Regardless of the concept that the CVZ mice would have greater capacity to generate microbial EAA, it would be important to control for the low and high protein diets across both GF and CVZ mice and especially for interpretation and context for the 16S rRNA sequencing data. For example, the 16S rRNA sequencing data would be more meaningful if microbial composition was compared between a low protein diet and a high protein diet (i.e., which phyla or genera shift in abundance in response to adequate or low protein diets)? Which microbes are likely facilitating EAA production? Metagenomic and metatranscriptomic analyses would also help in this regard.
2. Was a regional assessment of microbial composition considered? Small bowel vs large bowel vs stool? This may have greater relevance to absorption in the small intestine as alluded to in the introduction.

Minor Comments

1. Lines 24-27: the experimental design is confusing as described. How many treatment groups exactly?
2. Figure 1: Hypothesis section – unclear what the circles and triangles represent (circles for Conv?)
3. Lines 47-51: Are EAA producing microbes found in the small intestine? And what are they? This would be helpful information to include in the introduction.
4. Line 115: typo in the sentence “The mice were reared in...” and recommended to check document for other instances (see line 253).
5. Was a culture-based method employed to verify GF mice remained GF and that the Streptococcaceae ASVs were not from live bacteria and possibly from the food as suggested?
6. 1400 reads per sample is quite low.
7. Lines 294-295: 30 days should be sufficient but might also be dependent on the recipient diet.

Reviewer #2

(Remarks to the Author)

The authors examine whether the gut microbiome provides essential amino acids (EAAs) to the host, using a germ-free vs. conventionalized mouse model. The authors use GF mice and FMT-CVZ mice fed isocaloric diets differing in protein content to create a protein-restricted scenario. They analyze stable carbon isotope signatures of six EAAs in multiple organs (brain, liver, kidney, muscle) to detect microbial contribution. In their findings, GF and CVZ mice showed nearly identical EAA isotopic profiles in tissues. Microbiome sequencing confirmed that CVZ mice had a normal gut bacterial community (dominated by Firmicutes and Bacteroidetes), while GF mice remained sterile. The results suggest that under these conditions, a reconstituted microbiome did not measurably contribute EAAs to the host. This is an important finding that challenges prior estimates of substantial microbial EAA provisioning, and it raises questions about the functional capacity of

FMT-derived microbiomes. The study's approach (combining gnotobiotic mice with in situ isotope fingerprinting) is novel and provides a valuable, controlled test of microbial nutrient contribution. Overall, the work addresses a relevant nutritional and microbiome question, and the data are clearly presented. However, several points require clarification and further discussion.

1. The use of different protein levels for GF vs. CVZ mice is understood to test microbial rescue of EAA under protein restriction. However, this design also introduces a potential confounder, as dietary protein content differs between groups. The authors should more explicitly justify and discuss this choice. In particular, were GF mice given 18% protein solely to prevent deficiency, and was 10% protein for CVZ chosen based on prior studies showing microbial contributions under non-deficient restriction? It would strengthen the paper to acknowledge that the GF and CVZ groups had inherently different nutritional inputs, and to clarify that any differences in EAA profiles would be attributed to microbial action rather than diet per se.
2. A central question is whether the FMT-reconstituted microbiome fully recapitulates a natural, complex microbiota's function. The authors raise the possibility that the lack of detectable EAA contribution may reflect incomplete functional restoration in the CVZ mice. This point should be expanded. For instance, what was the diversity and composition of the transplanted microbiome compared to a typical specific-pathogen-free mouse? The 16S profiling (Figure 4) shows dominant taxa at family/genus level, but did the CVZ community achieve normal diversity/richness? If known EAA-synthesizing taxa (e.g., certain Bacteroides or Firmicutes capable of amino acid biosynthesis) were absent or at low abundance in these mice, that could explain the outcome. The authors are encouraged to comment on whether any specific microbial functions might have been missing in the reconstituted community.
3. The conclusion of "no evidence for microbial EAA contribution" relies on the absence of significant isotope signature differences between GF and CVZ mice. The authors should provide more details on the sensitivity of this isotopic approach. What is the threshold of detection for a microbial contribution using the fingerprinting?
4. The manuscript nicely reviews prior work, but the Discussion could better reconcile the contrasting results. The present study, with proper GF controls, finds essentially 0%. This is a striking discrepancy that should be explicitly discussed. The authors have an opportunity to clarify why their controlled approach yields a different conclusion. For example, were previous natural-abundance studies possibly confounded by unaccounted diet-tissue fractionation or other metabolic routing that mimicked a microbial signature?

Version 1:

Reviewer comments:

Reviewer #1

(Remarks to the Author)

Overall, the authors have nicely addressed the previous concerns raised by reviewers. The introduction has been much improved, easy to read and explains the EAA-fingerprinting method clearly.

Remaining minor concerns:

Line 141: typo "addresses" should be "address"

Line 144: suggested to remove the word "possibly"

Reviewer #2

(Remarks to the Author)

The authors have addressed all my comments.

Reviewer #1 (Remarks to the Author):

Summary

The goal of this work was to determine the microbial contribution of EAA to the host and distribution across several host tissues by comparing conventionalized mice on a low protein diet and germ free (GF mice) on a high protein diet. Microbe-derived EAAs among host tissues or that differed from GF mice were not detected. The authors correctly point out that further studies utilizing varied dietary protein levels and additional analytical and -omic techniques, for example, are needed.

Major Comments

1. Regardless of the concept that the CVZ mice would have greater capacity to generate microbial EAA, it would be important to control for the low and high protein diets across both GF and CVZ mice and especially for interpretation and context for the 16S rRNA sequencing data. For example, the 16S rRNA sequencing data would be more meaningful if microbial composition was compared between a low protein diet and a high protein diet (i.e., which phyla or genera shift in abundance in response to adequate or low protein diets)? Which microbes are likely facilitating EAA production? Metagenomic and metatranscriptomic analyses would also help in this regard.

>>> We thank the reviewer for highlighting the importance of dietary controls and multi-omic integration. Below, we address these points:

Our experimental design prioritized testing *microbial rescue under protein restriction*, a scenario where microbial EAA synthesis would be most physiologically relevant. Thus, CVZ mice received a low-protein diet (10% lactic acid casein) to mimic restriction, while GF mice received an adequate-protein diet (18% lactic acid casein) to prevent deficiency. While it would be insightful comparing microbial communities across diets (low vs. high protein) in CVZ mice, it would not directly test microbial EAA provisioning *to the host*, which requires GF controls to isolate microbial effects. However, we agree that future work should explore diet-microbiome interactions more broadly (e.g., protein levels in both GF and CVZ groups).

The 16S rRNA data (**Fig. 6**) confirmed that CVZ mice harbored a community dominated by Firmicutes and Bacteroidetes, consistent with healthy mice. While metagenomic/metatranscriptomic analyses could identify taxa with EAA biosynthesis pathways, such data cannot quantify *host assimilation* of microbial EAAs. Our isotopic approach directly tests whether microbial metabolites are routed to host tissues, a critical gap in prior -omic studies. We fully agree that integrating -omics with isotopic tracing would strengthen mechanistic insights. In the revised Discussion (**lines 190–208**), we propose to expand experimental approaches to investigate the impact of various protein contents on the CVZ microbiome, pair d13C-EAA analysis with

proteomics to track microbial peptide incorporation, using gnotobiotic models with defined EAA-producing taxa.

2. Was a regional assessment of microbial composition considered? Small bowel vs large bowel vs stool? This may have greater relevance to absorption in the small intestine as alluded to in the introduction.

>>> We appreciate the reviewer's insight into the potential relevance of regional microbial sampling. Below, we clarify our rationale:

Fecal samples were used as a standardized proxy for gut microbiota, consistent with established practices in vertebrate models. While small intestine sampling could provide insights into the ileum vs colon EAA absorption, our isotopic approach focused on functional outcomes (host assimilation of microbial EAAs) rather than spatial microbial ecology.

The absence of isotopic evidence for microbial EAA contributions suggests that compartmentalized microbial specialization, if present, did not translate to detectable host nutritional input under our experimental conditions. This aligns with our revised introduction (**lines 43–51**), which clarifies that EAA absorption depends on microbial activity across the entire gut, not just the small intestine (which from the literature appear to be the major digestive region for EAA absorption).

Our goal was to test whether microbial EAAs are routed to host tissues, not to map regional microbial niches. Future studies could pair isotopic tracing with region-specific sampling (e.g., jejunal vs. colonic content) to resolve absorption mechanisms.

1. Lines 24-27: the experimental design is confusing as described. How many treatment groups exactly?

>>> We had GF mice (n =5) (control) and CVZ mice (n = 5) (treatment). We have clarified this sentence in the revised submission in **lines 280-292 and in revised conceptual figure 2**.

2. Figure 1: Hypothesis section d^p unclear what the circles and triangles represent (circles for Conv?)

>>> We acknowledge the lack of clarity in our previous Figure 1 (now **Figure 2**) legend and apologize for any confusion this may have caused. In the revised manuscript, we have moved the hypothesis section to a new figure (**Figure 1**) and clearly described the

symbols in the legend. We hope this revision makes clear that the figure and its legend are now unambiguous and aligns with the hypothesis and experimental design.

3. Lines 47-51: Are EAA producing microbes found in the small intestine? And what are they? This would be helpful information to include in the introduction.

>>> Current evidence indicates that there is no clear taxonomic distinction between EAA-producing and non-EAA-producing gut bacteria; many common gut microbes possess the genetic capacity for EAA biosynthesis, but this trait is not exclusive to specific genera or phyla. The main exception is obligate endosymbionts with highly reduced genomes, which often lack complete EAA biosynthetic pathways and rely on other, non-endosymbiotic gut bacteria with larger genomes to supplement these functions.

It is also important to note that the site of EAA absorption (such as the small intestine) does not necessarily correspond to the exclusive location of EAA-producing bacteria. While the small intestine is a primary site for amino acid absorption, EAA produced by microbes elsewhere in the gut - particularly in the large intestine, where microbial biomass is highest and food residence time is longer - can also contribute to the host amino acid pool. Amino acids absorbed in the large intestine are typically derived from both bacterial metabolism and endogenous sources (<https://doi.org/10.3945/jn.117.248187>).

We have clarified these points in the revised introduction (**lines 37-50**) to provide a more accurate context for the distribution and function of EAA-producing microbes in the gut.

4. Line 115: typo in the sentence ÉC;The mice were reared in,ÉD; and recommended to check document for other instances (see line 253).

>>> This has been corrected.

5. Was a culture-based method employed to verify GF mice remained GF and that the Streptococcaceae ASVs were not from live bacteria and possibly from the food as suggested?

>>> The gnotobiotic facility at which the experiments were carried out have an established protocol to verify and confirm the GF nature of the mice. These have consistently not yielded any bacteria as the citation listed and as per personal communications with our co-author, Dr. Ramer-Tait, who oversees the operations of the facility. We have clarified this in the revised submission in **lines 279-292**.

6. 1400 reads per sample is quite low.

>>> The reviewer is right that 1400 reads per sample is relatively low for 16S rRNA sequencing data. Several factors likely contributed to this outcome, including the nature of the conventionalization process (oral gavage), the establishment and recovery of the microbiome, and the dietary modifications implemented for the CVZ group. Additionally, the sequencing depth was influenced by the specific criteria and parameters used during sequence analysis and quality filtering. While we acknowledge this limitation, 1,400 reads per sample was the yield available for our analyses in this study. **We show sufficient depth recovered from sequencing efforts to proceed in Supplementary figure 1.**

7. Lines 294-295: 30 days should be sufficient but might also be dependent on the recipient diet.

>>> The reviewer is right about the impact of the recipient diet on the gut microbiome establishment in the CVZ mice. We have mentioned the sufficiency of the reconstitution period, and the possible impact of the initial diet and the subsequent experimental diet in **lines 195-208 and 219-225** in the revised submission.

Reviewer #2 (Remarks to the Author):

The authors examine whether the gut microbiome provides essential amino acids (EAAs) to the host, using a germ-free vs. conventionalized mouse model. The authors use GF mice and FMT-CVZ mice fed isocaloric diets differing in protein content to create a protein-restricted scenario. They analyze stable carbon isotope signatures of six EAAs in multiple organs (brain, liver, kidney, muscle) to detect microbial contribution. In their findings, GF and CVZ mice showed nearly identical EAA isotopic profiles in tissues. Microbiome sequencing confirmed that CVZ mice had a normal gut bacterial community (dominated by Firmicutes and Bacteroidetes), while GF mice remained sterile. The results suggest that under these conditions, a reconstituted microbiome did not measurably contribute EAAs to the host. This is an important finding that challenges prior estimates of substantial microbial EAA provisioning, and it raises questions about the functional capacity of FMT-derived microbiomes. The study's approach (combining gnotobiotic mice with in situ isotope fingerprinting) is novel and provides a valuable, controlled test of microbial nutrient contribution. Overall, the work addresses a relevant nutritional and microbiome question, and the data are clearly presented. However, several points require clarification and further discussion.

>>> We appreciate greatly the comments provided by reviewer 2. Our primary motivation for this study was to empirically test the emerging hypothesis that the vertebrate gut microbiome can supply essential metabolites, such as EAAs, to the host under physiologically relevant conditions. By employing a controlled gnotobiotic mouse model and in situ stable isotope analysis, we aimed to directly quantify the extent of microbial EAA contribution to host tissues.

The finding that GF and CVZ mice exhibited nearly identical $\delta^{13}\text{C}$ values for EAAs across multiple organs was unexpected and challenges prevailing estimates of substantial microbial EAA provisioning in vertebrates. This result suggests that, at least under the specific dietary and experimental conditions tested, a reconstituted microbiome did not measurably supplement host EAA pools. We recognize that this outcome raises important questions about the functional capacity of FMT-derived microbiomes and the context-dependence of microbial nutrient contributions.

We are currently conducting additional studies with modified experimental designs to further investigate the factors that may microbiome reconstitution and influence microbial EAA provisioning, including variations in dietary protein levels, microbiome composition, and host-microbe interactions. We thank the reviewer for recognizing the novelty and rigor of our approach, and we look forward to addressing the specific points raised in the following sections.

1. The use of different protein levels for GF vs. CVZ mice is understood to test microbial rescue of EAA under protein restriction. However, this design also introduces a potential confounder, as dietary protein content differs between groups. The authors should more explicitly justify and discuss this choice. In particular, were GF mice given 18% protein solely to prevent deficiency, and was 10% protein for CVZ chosen based on prior studies showing microbial contributions under non-deficient restriction? It would strengthen the paper to acknowledge that the GF and CVZ groups had inherently different nutritional inputs, and to clarify that any differences in EAA profiles would be attributed to microbial action rather than diet per se.

>>> We thank the reviewer for highlighting the importance of dietary controls and the potential for confounding due to differing protein levels between GF and CVZ mice. We address these points below:

Our experimental design intentionally used different protein levels to test the hypothesis that the gut microbiome can rescue the host from EAA limitation under protein-restricted conditions. As we mentioned above in our reply to reviewer 1, GF mice were provided with an 18% protein diet to prevent deficiency and ensure their health and survival throughout the experiment, as GF animals are particularly sensitive to nutritional stress

and may not tolerate protein restriction without adverse effects. In contrast, CVZ mice received a 10% protein diet, which is sufficient for survival but low enough to create a scenario where microbial EAA provisioning could be physiologically relevant. The source of protein (casein) and all other dietary components were identical between groups, as detailed in Table 2, with the only variable being the amount of casein. The 10% protein level for CVZ mice was selected based on prior studies demonstrating that this level is sufficient to reveal potential microbial contributions to host EAA pools under non-deficient but restricted conditions.

We agree that this design results in inherently different nutritional inputs for the two groups, and we have clarified in the revised manuscript (**Table 2 heading**) that any observed differences in $\delta^{13}\text{C}$ -EAA profiles between GF and CVZ mice would be attributable to microbial action rather than dietary composition per se. The use of a higher protein diet in GF mice was not intended to confound the results, but was a necessary measure to avoid protein deficiency, which could introduce additional physiological variables.

2. A central question is whether the FMT-reconstituted microbiome fully recapitulates a natural, complex microbiota's function. The authors raise the possibility that the lack of detectable EAA contribution may reflect incomplete functional restoration in the CVZ mice. This point should be expanded. For instance, what was the diversity and composition of the transplanted microbiome compared to a typical specific-pathogen-free mouse? The 16S profiling (Figure 4) shows dominant taxa at family/genus level, but did the CVZ community achieve normal diversity/richness? If known EAA-synthesizing taxa (e.g., certain Bacteroides or Firmicutes capable of amino acid biosynthesis) were absent or at low abundance in these mice, that could explain the outcome. The authors are encouraged to comment on whether any specific microbial functions might have been missing in the reconstituted community.

>>> The CVZ mice exhibited a gut microbiota dominated by Firmicutes and Bacteroidetes (**Fig. 6**), consistent with the taxonomic profile of healthy, specific-pathogen-free (SPF) mice. While α -diversity metrics (e.g., Shannon index) are not absolute indicators of functional restoration, the observed community structure aligns with prior FMT studies in gnotobiotic models. Facility validation protocols confirmed that the reconstituted microbiome matched expected taxonomic profiles for SPF mice, as demonstrated in previous work at the Nebraska Gnotobiotic Mouse Program.

As mentioned above, EAA biosynthesis is a widespread metabolic capability among gut bacteria, not restricted to specific taxa. Genomic surveys indicate that most gut-associated bacteria (including Bacteroidetes, Firmicutes, and Proteobacteria) possess pathways for synthesizing essential amino acids.

The absence of isotopic evidence for microbial EAA contributions in the CVZ mice, despite having comparable microbiomes to other reconstituted mice microbiomes (**Fig. 6**) might be because microbial EAAs were either not produced in sufficient quantities (because protein was not sufficiently deficient), not absorbed, or routed to microbial biomass rather than host tissues.

However, while the transplanted microbiome taxonomically resembled a natural community, functional redundancy or metabolic prioritization (e.g., microbial self-use of EAAs) could explain the lack of detectable host assimilation. Additionally, we have emphasized the fact that the CVZ mice were maintained on a relatively high protein-diet during the gut microbiome establishment period, and that this may have selected for microbes adapted to high protein conditions, and thus, limited or reduced subsequent EAA production on the experimental low-protein diet. We have revised our discussion to incorporate this view as per the suggestion in **lines 219–225**), taking into consideration the diet-dependent nature of the reconstitution process in the CVZ.

3. The conclusion of no evidence for microbial EAA contribution relies on the absence of significant isotope signature differences between GF and CVZ mice. The authors should provide more details on the sensitivity of this isotopic approach. What is the threshold of detection for a microbial contribution using the fingerprinting?

>>> The general absence of significant $\delta^{13}\text{C}$ -EAA offsets ($\Delta\delta^{13}\text{C} = \delta^{13}\text{C}_{\text{Host-EAA}} - \delta^{13}\text{C}_{\text{Diet-EAA}}$) between germ-free (GF) and conventionalized (CVZ) mice constitutes the primary evidence against microbial EAA provisioning. Under our experimental conditions, the maximal observed offset in CVZ mice was 0.6‰ (muscle tissue), below the empirically validated detection threshold of $\geq 1\%$ (<https://doi.org/10.1111/j.1365-2656.2010.01722.x>). This threshold accounts for analytical error ($\pm 0.3\%$) and biological variability, ensuring robust sensitivity to microbial contributions $\geq 5\text{--}10\%$ of total host EAA pools.

The $\delta^{13}\text{C}$ -EAA fingerprinting (linear discriminant analysis, LDA) served as a secondary, confirmatory tool to visualize isotopic equilibration to dietary proteins across organs and test for putative non-dietary EAA sourcing. By training the LDA model on reference $\delta^{13}\text{C}$ -EAA patterns from bacteria, fungi, plants, and dietary casein, we confirmed that host tissues equilibrated towards dietary casein, not bacteria (**Fig. 3**). This auxiliary analysis reinforced the primary isotopic evidence by demonstrating no detectable microbial "fingerprint" in any organ. This dual approach ensured both quantitative rigor (via thresholds) and spatial resolution (via multivariate analysis).

In the revised text (**lines 50–70**), we:

- Explicitly define the $\Delta\delta^{13}\text{C}$ detection threshold and its validation.
- Detail the LDA's role as a supplementary visualization tool.

- Emphasize that isotopic offsets, not fingerprinting alone, drive the conclusion.

4. The manuscript nicely reviews prior work, but the Discussion could better reconcile the contrasting results. The present study, with proper GF controls, finds essentially 0%. This is a striking discrepancy that should be explicitly discussed. The authors have an opportunity to clarify why their controlled approach yields a different conclusion. For example, were previous natural-abundance studies possibly confounded by unaccounted diet-tissue fractionation or other metabolic routing that mimicked a microbial signature?

>>> To address the discrepancy between our findings and prior natural-abundance studies, we have expanded the Discussion to explicitly consider methodological differences and potential confounding factors:

- **Lines 169-177 and 225-231:** We list the use of a GF control and the methodological and comparative approach this offered over other studies
- **Lines 178-196 and 245-249:** We discussed how the reconstituted gut microbiomes was like other reconstituted mouse gut microbiomes, listed the functionality of the major bacterial families, and commented on why we did not see a significant EAA contribution.
- **Lines 197-211 and 214-216:** We further discussed how the experimental design may have masked any significant detection EAA contribution. We commented on how the CVZ mice were fed a high-protein standard lab diet that was different in terms of protein content and protein sources from the actual OpenStandard diet during the experimental feeding period. This was done to avoid mortality of the newly gavaged mice but could have masked any significant detection.

Overall, in the Discussion, we do convey the sentiment that without GF controls, it is challenging to disentangle true microbial input from these background processes. Our approach eliminates some ambiguities from previous studies and provides a more accurate assessment of microbial EAA provisioning (we also eliminated coprophagy). However, we do raise the possibility of the diet-dependent nature of the reconstitution process and subsequent functionalities in the CVZ mice relative to mice with normal/natural microbiome and how this may have masked detection in this study. We stress that given the novelty of this approach, this study provides a glimpse into the experimental designs to use and the pitfalls to consider when investigating vertebrate host-microbiome interactions.